# Influence of Five Drying Methods on Active Compound Contents and Bioactivities of Fresh Flowers from *Syringa pubescens* Turcz

**DOI:** 10.3390/molecules28237803

**Published:** 2023-11-27

**Authors:** Weidong Xu, Jiameng Zhang, Yanfang Wu, Zichen Zhang, Xinsheng Wang, Junying Ma

**Affiliations:** 1College of Chemistry and Chemical Engineering, Henan University of Science and Technology, Luoyang 471023, China; xuweidongming@163.com (W.X.); zhang_jm1999@163.com (J.Z.); zhangzichener@163.com (Z.Z.); 2College of Basic Medicine and Forensic Medicine, Henan University of Science and Technology, Luoyang 471023, China; wyf245@163.com

**Keywords:** *Syringa pubescens* Turcz., drying method, bioactive component, biological activity

## Abstract

The flower of *Syringa pubescens* Turcz. is used in Chinese folk medicine and also as a flower tea for healthcare. The effects of five drying methods on the active compound contents, the antioxidant abilities, anti-inflammatory properties and enzyme inhibitory activities were evaluated. The plant materials were treated using shade-drying, microwave-drying, sun-drying, infrared-drying and oven-drying. The seven active compounds were simultaneously determined using an HPLC method. Furthermore, the chemical profile was assessed using scanning electron microscopy, ultraviolet spectroscopy and infrared spectroscopy. The antioxidant capacities and protective effects on L02 cells induced with hydrogen peroxide were measured. The anti-inflammatory effects on lipopolysaccharide-induced RAW264.7 cells were investigated. The enzyme inhibitory activities were determined against α-amylase, α-glucosidase cholinesterases and tyrosinase. The results indicated that drying methods had significant influences on the active compound contents and biological properties. Compared with other samples, the OD samples possessed low IC_50_ values with 0.118 ± 0.004 mg/mL for DPPH radical, 1.538 ± 0.0972 for hydroxyl radical and 0.886 ± 0.199 mg/mL for superoxide radical, while the SHD samples had stronger reducing power compared with other samples. The SHD samples could be effective against H_2_O_2_-induced injury on L02 cells by the promoting of T-AOC, GSH-PX, SOD and CAT activities and the reducing of MDA content compared with other samples. Furthermore, SPF samples, especially the SHD sample, could evidently ameliorate inflammation through the inhibition of IL-6, IL-1β and TNF-α expression. All the studied SPF samples exhibited evidently inhibitory effects on the four enzymes. The IC_50_ values of inhibitory activity on α-glucosidase and α-amylase from SHD sample were 2.516 ± 0.024 and 0.734 ± 0.034 mg/mL, respectively. SD samples had potential inhibitory effects on cholinesterases and tyrosinase with IC_50_ values of 3.443 ± 0.060 and 1.732 ± 0.058 mg/mL. In consideration of active compound contents and biological activities, it was recommended that SHD and SD be applied for drying SPF at an industrial scale.

## 1. Introduction

*Syringa pubescens* Turcz. (*S. pubescens*), which is a deciduous shrub, is widely distributed in mountainous areas of the Henan, Shanxi, Shaanxi, Hebei and Shandong Provinces and Beijing City [1]. The dried flower of *Syringa pubescens* (Syringae Pubescentem Flos, SPF) is often used as a Chinese folk medicine and flower tea for healthcare in the Funiu Mountains in Henan Province. To date, modern scientific investigations on SPF have indicated that it has diverse biological and pharmacological properties, including antioxidant [2,3], hepatoprotective [4], anti-inflammatory and antibacterial [5] activities. Furthermore, our recent study has demonstrated that SPF possessed strong inhibitory activity against α-glucosidase [1]. These biological properties are due to the presence of chemical constituents in SPF. The main compounds in SPF are phenylethanoid glycosides and iridoids [1,2,3,6,7].

The drying method of medicinal plants is one of the key steps affecting their compounds and biological activities [8,9,10]. Among the various drying methods, shade-drying (SHD), sun-drying (SD) and oven-drying (OD) are commonly used to process medical plants at industrial scales [11,12]. SHD and SD have successfully long been used for the drying of plant material. However, the two drying methods have disadvantages, including the slowness of the process, exposure to environmental contamination, uncertainty of weather and high labor requirements. Compared with SHD and SD, OD could be used for the drying of most types of plant materials cost- and time-effectively. In recent years, with studies conducted, novel drying methods, including microwave-drying (MD) and infrared-drying (IRD), have been developed for the drying of plant materials [13,14]. It has been proved that MD could offer a higher drying rate and shorter drying time, higher rehydration ratio and lower shrinkage. IRD has the many benefits of uniform heating, low processing time, high heat transfer rate and good energy consumption. A different drying method is suitable for different medicinal herbals because the drying process can lead to the loss of heat-labile compounds. However, the drying method of SPF and the differences in the main bioactive compounds during the different drying methods have not been studied. Therefore, the aim of this present work was to explore the influences of changes in drying methods on the bioactive compound contents, the antioxidant capacities, anti-inflammatory properties and enzyme inhibition effects of SPF to determine the most suitable drying method.

## 2. Results and Discussion

### 2.1. The Contents of Bioactive Compounds

The contents of the bioactive compounds of SPF prepared using five drying methods are shown in Table 1. The data revealed that echinacoisde, forsythoside B, verbascoside and oleuropein (Figure 1) were the major compounds of the SPF extract, which was in agreement with a previous study [1]. Meanwhile, it could be found that the contents of the active ingredients exhibited significant differences under the five drying methods (*p* < 0.05). The content of salidroside was the lowest content in the SD method. The maximal ayringin was observed in SD (0.317 ± 0.040 mg/g) > OD and SHD (0.161 ± 0.005, 0.149 ± 0.015 mg/g) > MD (0.028 ± 0.004 mg/g) and the minimal from IRD (0.018 ± 0.007 mg/g). As for echinacoside, the contents from different drying methods were significantly different (*p* < 0.05). The highest content of echinacoside was found in SD and the minimal in IRD. The contents of forsythoside B and verbascoside showed similar trends, and the concentrations in OD were higher than those in other drying methods. The oleuropein content was MD > SD > OD and SHD > IRD. It could be concluded that the drying of SPF should not be performed using IRD, possibly due to the fact that IRD could cause degradation in the active compounds of SPF [15]. Based on the contents of bioactive compounds, MD, SD, OD and SHD could be used to treat SPF. Meanwhile, MD was the fastest method among these drying processes. However, SD, OD and SHD have been considered to be the most prevalent drying methods for industrial production [16]. In consideration of energy savings, SD and SHD are the appropriate drying methods for SPF.

### 2.2. Scanning Electron Microscopy (SEM) Analysis

The powder characterization of SPF was used to assess the differences in samples treated using five drying methods. Figure 2a–e shows the micromorphology of different samples. It could be seen that powder features exhibited a rough and irregular surface. Meanwhile, it was evident that the micromorphological characteristics of samples treated with different drying methods possessed significant differences. The reason was possibly attributed to the fact that the five drying methods resulted in physical and chemical changes in the SPF materials [17].

### 2.3. FT-IR Spectra Analysis

The intensity, shape and position of FT-IR spectra peaks from samples prepared with five drying methods were used to compare the differences and similarities (Figure 3). It could be observed that the chemical profiles of SPF samples dried by different methods were similar owing to the similarity of the FT-IR spectra. The characteristic absorption peaks of glycosides, including echinacoisde, forsythoside B, verbascoside and oleuropein, could be found in the FT-IR spectra. On the other hand, differences in the positions and intensities in the FT-IR spectra were found in the 1700–850 cm^−1^ wavenumber regions. The changes in position and intensity of the absorption peaks revealed the differences in component contents of samples prepared by the five drying methods. Compared with other drying methods, the absorption bands of samples treated with IRD at 1634–1632 cm^−1^, 1517–1512 cm^−1^, 1382–1378 cm^−1^, 1263–1256 cm^−1^, 1157–1152 cm^−1^, 1074–1066 cm^−1^, 927 cm^−1^ and 860–858 cm^−1^ were low, suggesting that the active compounds might be degraded after the IRD process.

### 2.4. UV Spectroscopy Analysis

The UV-Vis spectra of the ethanol extract of five samples are illustrated in Figure 4. The peaks at 285 and 332 nm can be associated with the presence of oleuropein and phenylethanoid glycosides, respectively [18,19]. It could be observed that the bands associated with the major compounds of SPF extract after different drying methods were similar. However, there were some variations in the absorbance values of samples prepared using the different drying methods.

### 2.5. Antioxidant Activities In Vitro

#### 2.5.1. Influence of Five Drying Methods on Reducing Power

The reducing power of the SPF samples is shown in Figure 5a,b. Figure 5a shows that the reducing power of the SPF samples from the five drying methods was lower than that of the VC in the testing concentration range. Meanwhile, compared with other drying methods, the reducing capacity of the sample after IRD was weaker. The reducing capacities of the samples prepared using SD, SHD, MD and OD exhibited similar trends and were linear within the concentration from 0 to 3.25 mg/mL. When the concentration was more than 3.25 mg/mL, the reducing power did not increase. Likewise, the results of FRAP confirmed that SPF samples prepared using the five drying methods possessed different antioxidant properties (Figure 5b); meanwhile, the sample after SHD had the highest antioxidant effect. However, the SPF sample exhibited stronger antioxidant properties compared to the Fe(III) reducing power method. Furthermore, the antioxidant capacity of the SPF sample was closer to that of the VC at a high concentration (4 mg/mL).

#### 2.5.2. Influence of Different Drying Methods against DPPH, ABTS^+^ and •OH Scavenging Ability

In the present work, three free radicals, including DPPH, ABTS^+^ and •OH, were used to investigate the scavenging capacity of the SPF sample. Figure 5c shows the DPPH free radical scavenging effects of different SPF samples. It was evident that the DPPH radical scavenging capacity of VC was significantly stronger than that of the SPF sample within the test concentration (*p* < 0.05). The IC_50_ values of samples prepared with SHD, OD, MD, SD and IRD were 0.131 ± 0.002, 0.118 ± 0.004, 0.144 ± 0.004, 0.133 ± 0.004 and 0.142 ± 0.003 mg/mL, respectively, and the sample after OD possessed the strongest DPPH scavenging effect. The ABTS^+^ radical scavenging effects of the SPF sample had similar trends to those of the DPPH radical scavenging effects (Figure 5d). The IC_50_ values of corresponding samples were 0.170 ± 0.010, 0.175 ± 0.007, 0.190 ± 0.008, 0.174 ± 0.006 and 0.199 ± 0.006 mg/mL, respectively. It could be found that the IC_50_ values of the ABTS^+^ radical scavenging effect were higher than those of the DPPH radical scavenging effect. The •OH scavenging effect of the SPF samples was similar to those found for ABTS^+^ and DPPH (Figure 5e). The IC_50_ values were 1.621 ± 0.018, 1.538 ± 0.097, 1.805 ± 0.088, 1.598 ± 0.091 and 2.100 ± 0.134 mg/mL, respectively. Meanwhile, the IC_50_ values of the •OH scavenging abilities were higher compared to ABTS^+^ and DPPH free radicals. Among the five drying methods, the samples after OD had the highest •OH scavenging effect. Based on scavenging ability against three free radicals, OD and SHD were suitable for the drying of SPF.

#### 2.5.3. Influence of Different Drying Methods on Superoxide Radical Scavenging Effect

The SPF sample against the superoxide radical scavenging activity is shown in Figure 5f. In the test concentration range of 0.25–6.0 mg/mL, the superoxide radical scavenging abilities of SPF prepared with five drying methods were linear. Compared to the other SPF samples, the SHD sample possessed a stronger superoxide radical scavenging capacity. Nevertheless, the activity of the SPF sample was weaker than that of the VC.

### 2.6. Protective Effect of SPF Extract on H_2_O_2_-Induced Oxidative Injury in L02 Cells

#### 2.6.1. SPF of Cytotoxicity on L02 Cells

The cytotoxicity of the SPF extract is shown in Figure 6a. It could be found that the cell viabilities treated by the SPF extract were more than 95% at the tested concentration (37.5 μg/mL, 75 μg/mL and 150 μg/mL), indicating that the SPF extract had noncytotoxicity on L02 cells [20]. Therefore, the SPF extract concentrations of 37.5 μg/mL, 75 μg/mL and 150 μg/mL were used in the following investigation.

#### 2.6.2. Protective Effect of SPF Extract against Oxidative Damage by H_2_O_2_-Induced Cells and Determination of Cells’ Biochemical Indexes

The protective effect of the SPF extract on oxidative damage by H_2_O_2_-induced cells is displayed in Figure 6b. Compared to the control group, the cell survival rate induced by H_2_O_2_ significantly decreased at 150 μmol/L (*p* < 0.01). The cell viability notably increased after treatment with the SPF extract (75 μg/mL and 150 μg/mL) compared with the H_2_O_2_ group. Meanwhile, the cell viability of the SHD and SD samples exhibited an obvious increase at the concentration of 37.5 μg/mL (*p* < 0.01).

The results of SPF on the levels of MDA, T-AOC, SOD, CAT and GSH-PX in the cells are shown in Figure 6c. It could be found that the MDA level in the H_2_O_2_ group significantly increased compared with the control group (*p* < 0.01) and obviously decreased in the SPF groups (150 μg/mL) compared with the H_2_O_2_ group (*p* < 0.01). Furthermore, the MDA level treated with SPF prepared with SHD was lower than that of the other SPF groups. The antioxidant enzymes activities, including T-AOC, SOD, CAT and GSH-PX in the H_2_O_2_ group, significantly decreased compared with control group (*p* < 0.01). The antioxidant enzyme activities in the SPF groups (150 μg/mL) considerably increased compared with the H_2_O_2_ group (*p* < 0.01). Meanwhile, the increase trends in antioxidant enzyme activities were similar. As for T-AOC, the SHD group had relatively strong enzyme activity, followed by the SD group, OD group, MD group and IRD group. The findings obtained from this study were consistent with previous studies that the cells exposed to excessive hydrogen peroxide, the enzymatic activities were remarkably decreased together with an increase in the MDA level [21]. These results demonstrated that SPF could ameliorate oxidative damage by H_2_O_2_-induced L02 cells.

#### 2.6.3. Intracellular ROS Evaluation

The levels of intracellular ROS and ROS scavenging abilities of SPF are given in Figure 7a–h. The highest intracellular ROS level was found in the H_2_O_2_ group among the test groups (Figure 6c). Compared to the H_2_O_2_ group, the intracellular ROS level in the SPF groups significantly decreased (*p* < 0.05). Meanwhile, the intracellular ROS level in the SHD group was the lowest (Figure 7a,b). These results indicated that SPF could scavenge the intracellular ROS. Similar results were obtained by Shen et al. [22], who confirmed that phenylethanoid glycosides from *Incarvillea compacta* possessed intracellular ROS scavenging capacities.

### 2.7. Anti-Inflammatory Activity of SPF

#### 2.7.1. Cytotoxicity of SPF

The cytotoxicity of SPF extract on RAW264.7 cells is shown in Figure 8a. Compared with the control group, the viability of RAW264.7 cells exceeded 90%, suggesting SPF extract (37.5, 75, 150 μg/mL) showed non-cytotoxicity to RAW264.7 cells.

#### 2.7.2. Determination of NO, IL-6, IL-1β, and TNF-α of RAW.264.7 Cells

The NO is an inflammatory mediator and plays a vital role in the pathogenesis of inflammation [23]. The results are shown in Figure 8b. Compared with the control group, the NO production significantly increased (*p* < 0.01). However, NO production in the SPF groups was remarkably suppressed compared with the LPS group. Meanwhile, the SPF samples prepared with SHD and SD strongly possessed inhibition on the NO production.

The inflammation process could lead to an overexpression of inflammatory cytokines, including IL-6, IL-1β and TNF-α [24,25]. These inflammatory factors can promote the development of the degree of inflammation. It can be found in Figure 8c–e that the levels of IL-6, IL-1β, and TNF-α in RAW264.7 cells stimulated with LPS notably increased compared with the control group (*p* < 0.01). However, the levels of the inflammatory factors in the SPF groups significantly reduced compared with the LPS group. Furthermore, the SPF sample prepared with SHD could evidently inhibit the expression of IL-6, IL-1β and TNF-α.

### 2.8. Enzyme Inhibitory Activity

The α-amylase and α-glucosidase are the key enzymes for hydrolyzing carbohydrates into glucose [26], which causes an increase in postprandial blood glucose. The inhibition activities on α-amylase and α-glucosidase of the SPF extracts are presented in Figure 9a,b. It could be observed that the SPF extract exhibited enzyme properties in a concentration-dependent manner. Meanwhile, the IC_50_ value of the sample prepared with SHD was lower than that of other samples. However, the IC_50_ values of all tested SPF samples were higher than that of the positive reference (Acarbose, Appendix A). These results were compatible with our previous work in that the different original samples possessed α-glucosidase inhibition activities [1].

ROS have been confirmed to be strongly correlated with neurodegenerative diseases such as Alzheimer’s disease and Parkinson’s disease [27,28]. Previous studies have revealed that inflammation is involved in Alzheimer’s disease and Parkinson’s disease [29,30]. The SPF extract possessed strong antioxidant and anti-inflammatory activities [6,7]. The inhibition of AChE has been considered the main approach for the treatment of Alzheimer’s disease [31]. Moreover, studies have indicated that the inhibition of tyrosinase is a vital target in developing new drugs for Parkinson’s disease [32]. In this study, the enzyme inhibitory activities of SPF samples prepared with five drying methods were investigated. The results are presented in Figure 8c,d. It was evident that the SPF extracts possessed enzyme inhibitory activities in a concentrated manner. As for AChE inhibitory activity, the IC_50_ value of the sample after SD was lower than that of the other samples. Compared with all tested samples, the IC_50_ value of control was lower. Meanwhile, we also found a dose–response effect of tryosimase inhibitory activity of the SPF sample (Figure 8). It could be observed that the IC_50_ values (1.717 ± 0.188, 1.723 ± 0.058, Appendix A) of samples after SHD and SD were lower than that of other samples. The IC_50_ value of control (VC) was 0.033 ± 0.003, which was significantly lower than that of all tested SPF samples. The findings obtained from this work showed that SPF had strong AChE and tyrosimase inhibition degrees, and the samples prepared with SHD and SD possessed higher enzyme inhibitory capacities.

## 3. Materials and Methods

### 3.1. Plant Materials and Reagents

The flowers of *Syringa pubescens* Turcz. were obtained from Xin’an County, Henan Province, China, and were identified by Professor Yanfang Wu. Standard reference compounds, namely, salidroside, syringin, echinacoisde, forsythoside B, verbascoside, isoacteoside and oleuropein, were purchased from Chengdu Biopurify Phytochemicals Ltd. (Chengdu, China), and the purity was more than 98%. Additionally, 2,2′-Azinobis (3-ethylbenzothiazoline-6-sulfonic acid) diammonium salt (ABTS), 1,1-diphenyl-2-picrylhydrazyl (DPPH), 2,4,6-tris(2-pyridyl)-S-triazine (TPTZ), α-glucosidase (from baker’s yeast), and p-nitrophenyl-α-Dglucopyranoside (pNPG) were provided by Aladdin Chemical Co. (Shanghai, China). The α-amylase, cholinesterase and tyrosinase were obtained from Macklin (Shanghai, China). All other chemicals and solvents used were of analytical grade.

### 3.2. Drying Methods

The equal amounts of SPF were processed using five different drying methods. For the SD method, the fresh flower samples were exposed directly to the sunlight (22 °C) for 3 d at a well-ventilated place. For the SHD method, the fresh flower samples were placed under the shade in a laboratory for 1 week. The temperature of this laboratory was 25 °C with appropriate ventilation. In regards to the OD method, the process was conducted in a BPG-9106A drying oven (Yiheng Scientific Instrument Co., Shanghai, China) at 45 °C for 60 h. For the MD method, the fresh flower samples were spread on a hardened glass plate, which was placed into a power-adjustable microwave oven (700W in a PM20A0 microwave oven, Midea Group Co., Ltd., Guangzhou, China) and then were dried under the output power of 700 W with a frequency of 1100 MHz and 2 min durations. For the IRD method, the drying process was carried out in an infrared drying oven (Hangzhou Qiwei Instrument Co., Ltd., Hangzhou, China) for 30 min with a frequency range from 10 THz to 100 THz. All samples were processed with different drying methods to a constant weight and then were powdered using a mill and sieved through a 60-mesh sieve. The sample powder prepared by five drying methods was kept at 4 °C until use.

### 3.3. Preparation of the Standard Solution

A mixed stock solution, which contained 90.00 μg/mL of salidroside, 63.64 μg/mL of syringing, 86.36 μg/mL of echinacoisde, 86.36 μg/mL of forsythoside B, 94.55 μg/mL of verbascoside, 90.91 μg/mL of isoacteoside and 70.00 μg/mL of oleuropein, was prepared according to the previous study [1]. The solution was serially diluted in methanol to plot calibration curves.

### 3.4. Extraction of Bioactive Compounds of the SPF Samples

The extraction was conducted according to the previous study [1]. Briefly, 2 g of sample powder was accurately weighed and mixed with 20 mL of 70%. The extraction was conducted using a Ymnl-2008D ultrasonic probe extraction device (Nanjing Immanuel Instrument Equipment Co., Ltd., Nanjing, China) for 40 min at 40 °C. After extraction, the crude extract was subjected to centrifuge at 12,000 rpm for 15 min. The supernatant was transferred into a volumetric flask of 25 mL and diluted to volume using 70% ethanol.

### 3.5. Chromatographic Methods

The HPLC analysis was performed on an Eclassical 3200 HPLC system (Dalian Elite Analytical Instrument Co., Ltd., Dalian, China) equipped with an ultraviolet detector, an autosampler, a binary pump, and a column oven. The separation of active compounds was conducted with an Elite Supersil ODS2 column (4.6 mm × 250 mm, 5 μm, Dalian Elite Analytical Instrument Co., Ltd., Dalian, China). The chromatographic conditions were the same as those described by Wang et al. [1].

### 3.6. SEM

After drying treatments, the morphological feature of the sample was analyzed with an SEM (FlexSEM 1000, Hitachi, Tokyo, Japan). The sample powders were placed on conductive adhesive tape and sprayed with a very thin layer of gold for SEM analysis.

### 3.7. FT-IR Analysis

An IRTracer-100 spectrometer (Shimadzu, Japan) was employed to record the diffuse reflectance spectra of the SPF powder. Briefly, the KBr powder was added to each sample, and then the mixture was ground again and pressed into a tablet. The FTIR spectra were recorded in the range of 4000–400 cm^−1^ at a resolution of 4 cm^−1^ with 16 scans for each spectrum.

### 3.8. UV Spectroscopy

A UV-2700 spectrophotometer (Shimadzu, Kyoto, Japan) was employed to measure the absorbance of the ethanol extract within the wavelength range of 600–200 nm. The characteristic peaks were identified, and the peak values were recorded.

### 3.9. Assay of Antioxidant Activities

#### 3.9.1. Determination of Reducing Capacity

The measuring method of reducing power was performed in our previously published work [33]. The absorbance at 700 nm was determined as the reducing capacity. In all the antioxidant activity experiments, the Vitamin C (VC) was used as a positive reference.

#### 3.9.2. Assay of the Ferric Reducing Antioxidant Power (FRAP)

The reducing power was also evaluated using the FRAP method according to the study reported by Yang et al. [2]. In this study, the FeSO_4_ solution was employed to plot a linear regression equation as follows: Y = 5.8853x + 0.1395 (R^2^ = 0.999).

#### 3.9.3. DPPH and ABTS Radical Scavenging Assay

The DPPH and ABTS radical scavenging capacities of SPF extracts were performed with the method described by Wu et al. [34].

#### 3.9.4. Hydroxyl Radical (•OH) Scavenging Assay

The •OH scavenging ability was measured using the orthophenanthroline method [35] with a slight adjustment. Briefly, the mixed solution containing 1 mL sample, 1 mL phosphate buffer solution (0.2 M, pH 7.4), 1 mL orthophenanthroline (2.5 mM), 1 mL ferrous sulfate solution (2.5 mM) and 1 mL H_2_O_2_ was incubated for 60 min at 37 °C. Then, the absorbance value was recorded as *A_i_* at 532 nm. The absorbance value was recorded as *A*_0_ without a tested sample. The ·OH scavenging ability was obtained as following Equation (1).
•OH scavenging capacity (%) = (1 − *A_i_*/*A*_0_) × 100(1)

#### 3.9.5. Superoxide radical (O_2_^•−^) scavenging assay

The ability of SPF extract to scavenge O_2_^•−^ was evaluated with a pyrogallol auto-oxidation system [36] with a few adjustments. Briefly, the mixed solution containing 0.5 mL Tris–HCl buffer (pH 8.2, 50 mM) and 0.4 mL sample solution was kept for 10 min at 25 °C. Then 150 μL pyrogallol solution (3 mM) was added and incubated for 5 min at 25 °C. The mixed reaction system was ended after an added 150 μL HCl. The absorbance value of the mixed solution at 325 nm was recorded as *A_i_*. The absorbance value was recorded as *A*_0_ without a tested sample. The superoxide radical scavenging ability was calculated using Equation (1).

#### 3.9.6. Protective Effect of SPF Extract on H_2_O_2_-Induced Oxidative Injury in L02 Cells

##### SPF of Cytotoxicity on L02

Hepatic L02 cells were cultured as described by Wang et al. [37]. Briefly, the DMEM medium containing 10% FBS, penicillin (100 U/mL) and streptomycin (100 μg/mL) were used to culture L02 cells. Then, L02 cells were placed in a humidified incubator at 37 °C under 5% CO_2_ with continuous growth for 48 h. The obtained L02 cells were seeded at a density of 1.0 × 10^5^ cells/mL with 100 μL/well. The L02 cells were continuously incubated to an adherence, and the initial culture medium was removed. The L02 cells were pretreated using different concentrations of the SPF extract dissolved in DMSO, and DMEM and DMSO were used as references. The L02 cells were continued to be treated for an additional 24 h, the supernatants were removed, and PBS was used to wash the cells 3 times. The cell viability was measured using the MTT assay.

##### Protective Effect of SPF Extract against Oxidative Damage by H_2_O_2_-Induced Cells and Determination of Cells’ Biochemical Indexes

The protective effects of the five drying methods of the SPF extract against the H_2_O_2_-induced oxidative injury in hepatic L02 were measured using the method reported by Chen et al. [38]. Briefly, after the L02 cells were incubated in a 96-well plate at 37.0 °C for 24 h, different concentrations of SPF extract were added to a 96-well plate and treated for 12 h. Then, 100 μL of H_2_O_2_ (300 μmol/L) was transferred into a 96-well plate and incubated for 6 h to induce oxidative damage. MTT was added, and the absorbance value was read at 492 nm. Meanwhile, the cell culture media induced with H_2_O_2_ were collected to determine the biochemical indexes. The parameters, including SOD, GSH-PX, MDA, CAT and T-AOC, were measured using detection kits (Nanjing Jiancheng Bioengineering Institute, Nanjing, China) based on the manufacturers’ instructions.

##### Intracellular ROS Evaluation

The assay of intracellular ROS was conducted based on the previous study [38]. Briefly, the oxidative damage was generated according to the above-mentioned method. Each well was added with DCFH-DA (10 μM). After incubation for 1 h, the intracellular ROS levels were measured using a fluorescent microscope (Olympus IX73, Olympus Corporation, Tokyo, Japan). The cells induced with H_2_O_2_ were used as negative references.

### 3.10. Anti-Inflammatory Property of SPF in LPS-Activated RAW264.7 Cells

#### 3.10.1. Cell Culture and Cytotoxic Effect of SPF

RAW264.7 cells were provided by Procell Life Science & Technology Co., Ltd. (Wuhan, China). The DMEM containing 10% FBS and 1% streptomycin/penicillin was employed to culture RAW264.7 cells under 37 °C and 5% CO_2_ conditions. The cytotoxicity of the SPF extract was assessed using the method reported by Zhen et al. [39]. Briefly, RAW264.7 cells were incubated in 96-well plates (1 × 10^5^ cells/well). After 24 h, different concentrations of SPF extract (37.5, 75, 150 μg/mL) were added to the 96-well plates for another 24 h. Then, the MTT (20 μL) was added to each well, and the cell culture was kept for 4 h. The absorbance value of the cell culture was read at 570 nm to evaluate the cytotoxicity of the SPF extract.

#### 3.10.2. Determination of NO, IL-6, IL-1β and TNF-α of RAW264.7 Cells

The RAW.264.7 cells were transferred into a 6-well plate and treated for 24 h. Then, different concentrations of SPF extract were added to the test wells and cultured for 2 h. Finally, the LPS solution was added to induce inflammation models and kept for an additional 24 h. The cell culture media was obtained using centrifugation. The NO, IL-6, IL-10 and TNF-α were measured with ELISA kits based on the kit instructions.

### 3.11. Enzyme Inhibitory Activity

#### 3.11.1. α-Amylase Inhibitory Property Assay

The α-amylase inhibitory activity was determined using starch as a substrate, as reported previously [40], with slight adjustments. The testing sample solution (1 mL) and 0.5 mL of α-amylase solution dissolved in phosphate buffer were mixed and kept for 5 min at 37 °C. Then, the reaction system was activated by adding the starch solution (0.5 mL) and incubating for 5 min at 37 °C. The reaction was terminated by the addition of HCl (1mL, 1 M). Finally, the I_2_-KI solution (0.5 mL) was added to the reaction mixtures. The absorbance value was read as *A_i_* at 630 nm. The absorbance value was read as *A*_0_ without a tested sample. The acarbose was used as a positive reference. The α-amylase inhibition activity was calculated with the Equation (2).
Inhibitory activity (%) = [(*A*_0_ − *A_i_*)/*A*_0_] × 100(2)

#### 3.11.2. Inhibitory Activity of α-Glucosidase

The inhibitory activity on α-glucosidase was investigated based on our previous study [1]. Briefly, the reaction system containing 100 μL testing sample, 100 μL glutathione, 100 μL α-glucosidase solution dissolved in phosphate buffer (pH 6.8) and PNPG (50 μL) was treated for 15 min at 37 °C. After pretreatment, the mixed solution was terminated with the addition of 100 μL sodium carbonate (0.2 M). The absorbance value was read as *A_i_* at 405 nm. The absorbance value was read as *A*_0_ without α-glucosidase. The α-glucosidase inhibition activity was calculated with the Equation (2).

#### 3.11.3. Inhibitory Activity on Tyrosinase Assay

The inhibitory property against tyrosinase was measured, as in our previously published paper [41]. Briefly, the reaction system containing L-DOPA or L-tyrosine aqueous solution (50 μL, 2 mM) and SPF solution was treated at 37 °C for 10 min. Then, the tyrosinase (100 μg/mL) was mixed with the solution to activate the reaction. After 10 min, the absorbance value of the sample was read as *A_i_* at 475 nm. The absorbance value without the sample was read as *A_1_*. The absorbance value without tyrosinase was read as *A*_0_. The tyrosinase inhibitory activity was calculated with the Equation (3).
Inhibitory activity (%) = 100 − [(*A_i_* − *A*_0_)/(*A*_1_ − *A*_0_)] × 100(3)

#### 3.11.4. Cholinesterase Inhibitory Activity Assay

The inhibitory activity ChE was assessed using the previously reported method [42]. The reaction system containing different concentrations of SPF extract (50 μL), DTNB solution (125 μL), AChE or BChE solution dissolved in Tris-HCl buffer (pH 8.0) (25 μL) was incubated for 15 min at 25 °C. The absorbance value of the sample was read as *A_i_* at 405 nm, and without the enzyme read as *A*_0_. The ChE inhibitory activity was calculated with the Equation (2).

## 4. Conclusions

In this work, the influences of five drying methods on the active ingredient contents, antioxidant properties, anti-inflammatory activity and enzyme inhibition activity of SPF were investigated for the first time. As a result, the SHD treatment required the longest drying times, while the retention of bioactive compounds was relatively high. The MD treatment was fast, and the content of oleuropein was the highest, which might be attributed to the shorter treatment time. Meanwhile, drying methods could considerably influence biological activities. For antioxidant activity, SHD samples exhibited a strong reducing power and superoxide radical scavenging effect, and could be effective against H_2_O_2_-induced injury on L02 cells. Furthermore, the SHD samples could evidently alleviate inflammation. For the enzyme inhibition effect, the SHD samples could notably inhibit activity on α-glucosidase and α-amylase, while SD samples had potential inhibitory effects on cholinesterases and tyrosinase. On the whole, SHD and SD were suitable to dry the SPF because of high bioactive constituent contents, strong antioxidant activities, high anti-inflammatory properties and enzyme inhibitory activities.

## Figures and Tables

**Figure 1 molecules-28-07803-f001:**
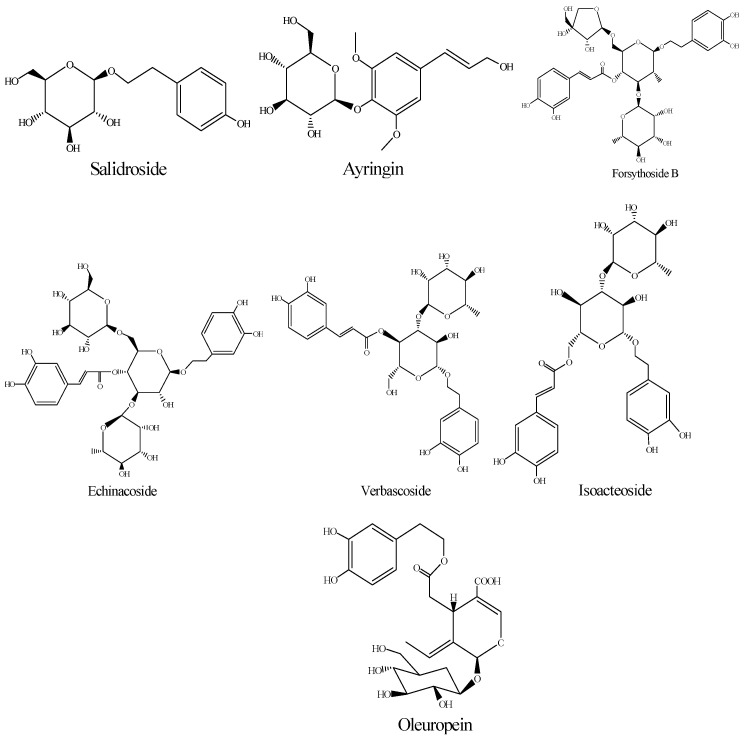
Chemical structure of bioactive compounds of SPF.

**Figure 2 molecules-28-07803-f002:**
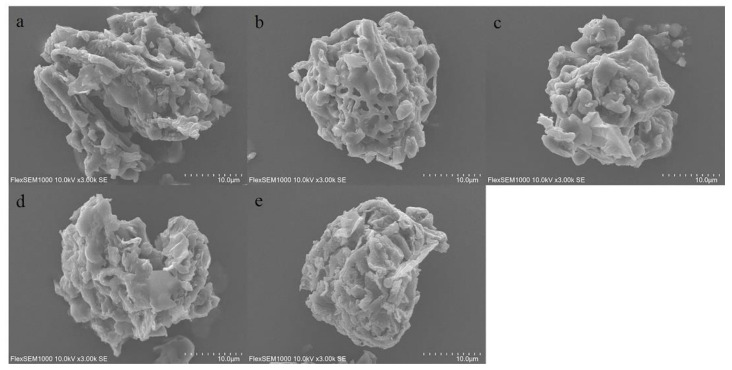
Scanning electron microscope images of SPF under different drying methods. Note: (**a**): SHD; (**b**): OD; (**c**): MD; (**d**): SD; (**e**): IRD.

**Figure 3 molecules-28-07803-f003:**
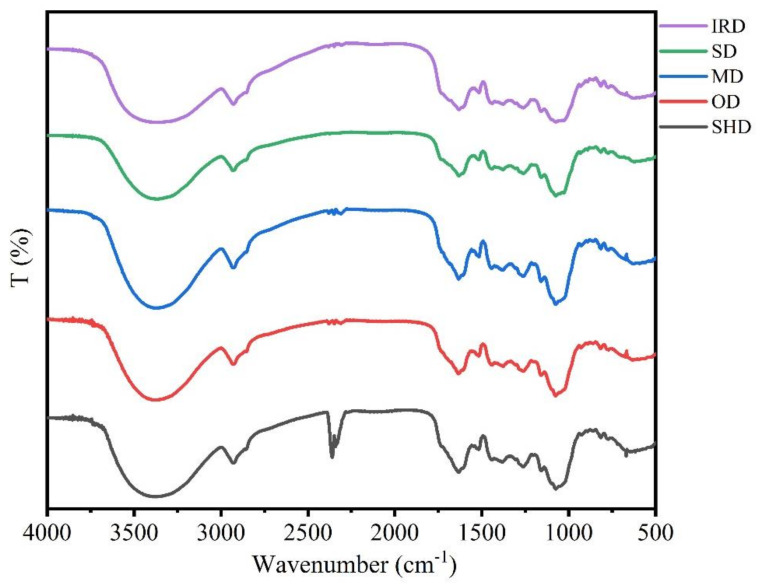
FT−IR spectra of SPF under different drying methods.

**Figure 4 molecules-28-07803-f004:**
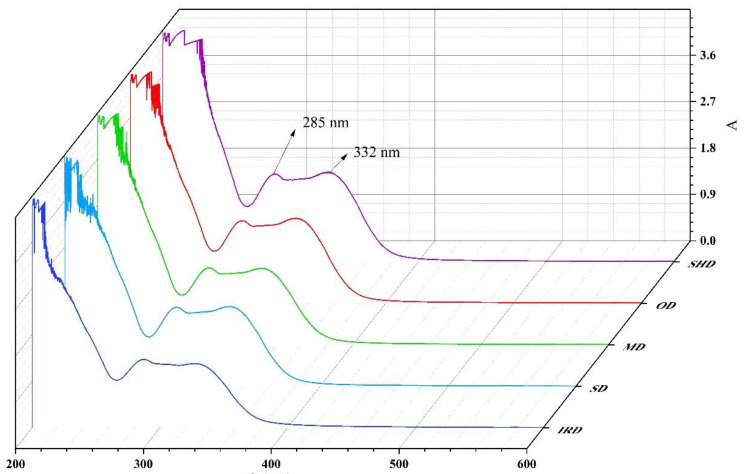
UV-Vis spectra of SPF under different drying methods.

**Figure 5 molecules-28-07803-f005:**
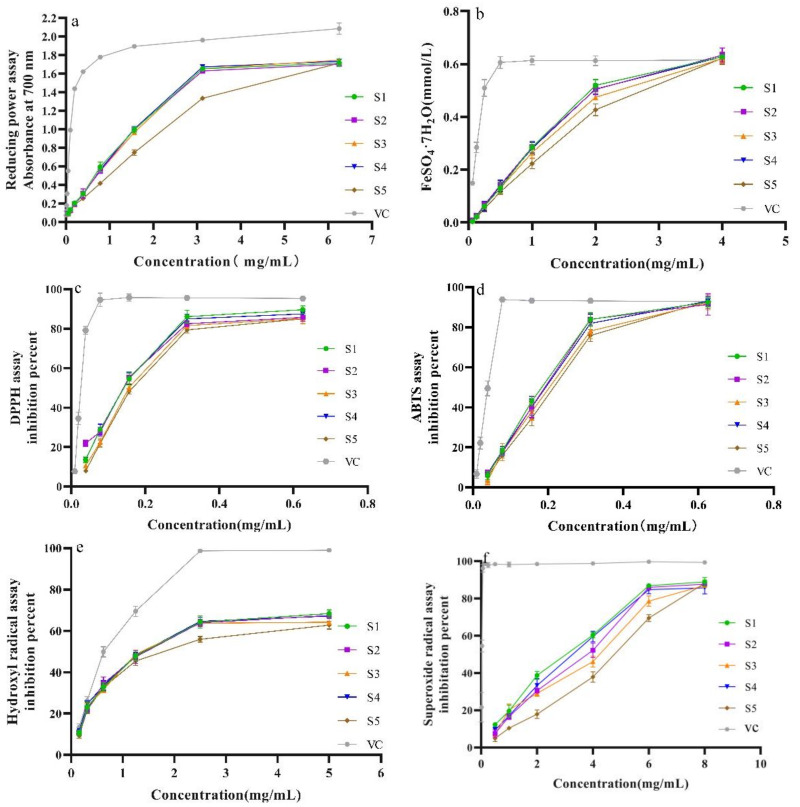
Antioxidant activities of SPF under different drying methods. (**a**) Reducing power, (**b**) FRAP, (**c**) DPPH, (**d**) ABTS, (**e**) •OH, (**f**) Superoxide radical. Note: S1: SHD; S2: OD; S3: MD; S4: SD; S5: IRD. (Similarly, hereinafter).

**Figure 6 molecules-28-07803-f006:**
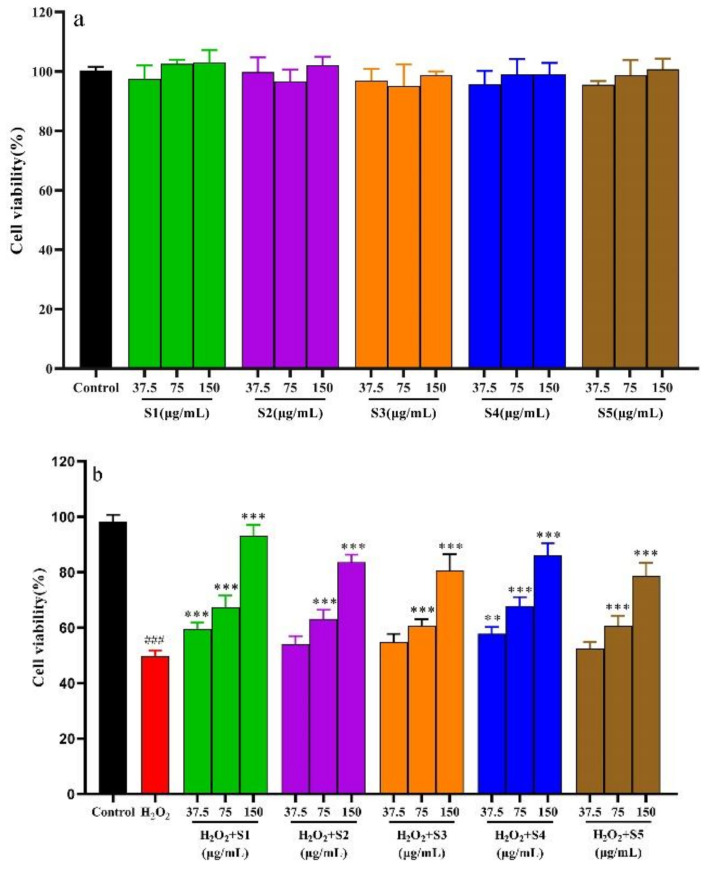
Effects of H_2_O_2_ and SPF on L02 cells. (**a**) Cytotoxicity of SPF; (**b**) Antioxidant activity of SPF against oxidative damage by H_2_O_2_-induced L02 cells; (**c**) Effects of SPF on biochemical indexes. * *p* < 0.05, ** *p* < 0.01 and *** *p* < 0.001 compared with Model group. ### *p* < 0.001 compared with Control group.

**Figure 7 molecules-28-07803-f007:**
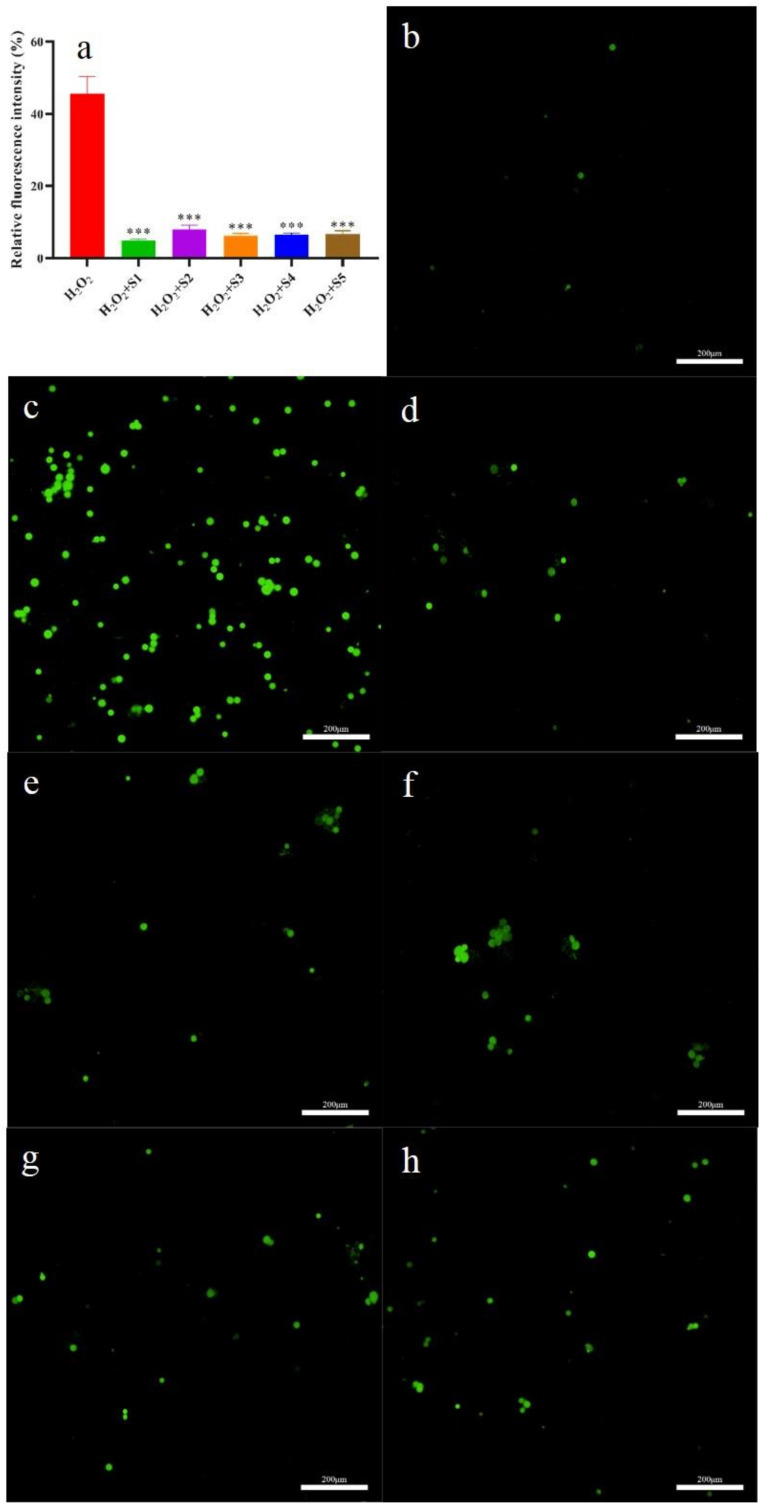
ROS levels of L02 cells treated using H_2_O_2_ and SPF. (**a**) Relative fluorescence intensity; (**b**) Control group; (**c**) H_2_O_2_ group; (**d**) H_2_O_2_ + S1 group; (**e**) H_2_O_2_ + S2 group; (**f**) H_2_O_2_ + S3 group; (**g**) H_2_O_2_ + S4 group; (**h**) H_2_O_2_ + S5 group. *** *p* < 0.001 compared with H_2_O_2_ group.

**Figure 8 molecules-28-07803-f008:**
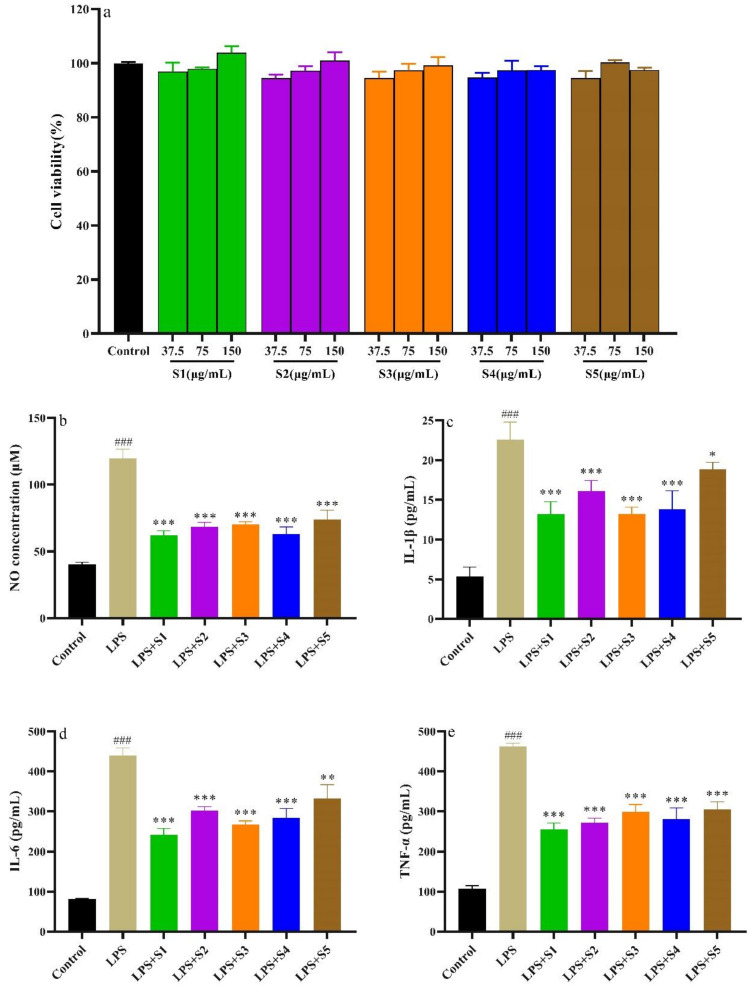
Effects of LPS and SPF on RAW264.7 cells. (**a**) Cytotoxicity of SPF; (**b**) Effects of SPF and LPS on NO; (**c**) Effects of SPF and LPS on IL-1β; (**d**) Effects of SPF and LPS on IL-6; (**e**) Effects of SPF and LPS on TNF-α. * *p* < 0.05, ** *p* < 0.01 and *** *p* < 0.001 compared with LPS group. ### *p* < 0.001 compared with Control group.

**Figure 9 molecules-28-07803-f009:**
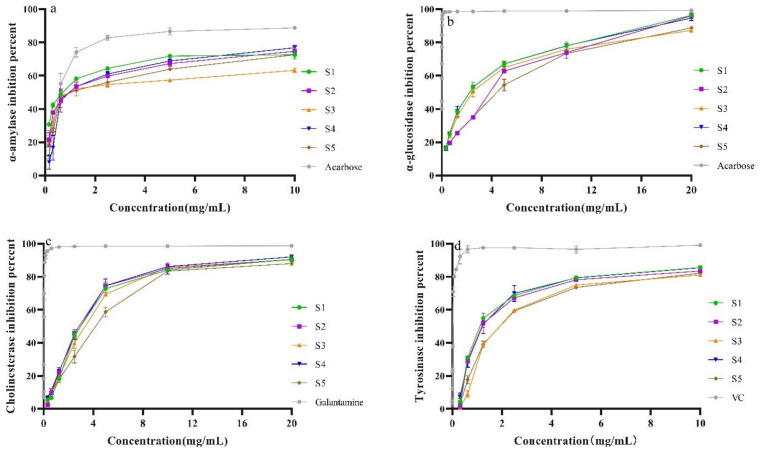
Enzyme inhibitory potentials. (**a**) α-amylase inhibitory activity of SPF; (**b**) α-glucosidase inhibitory activity of SPF; (**c**) AChE inhibitory activity of SPF; (**d**) Tyrosinase inhibitory activity of SPF.

**Table 1 molecules-28-07803-t001:** Effect of different drying methods on the contents of bioactive compounds in SPF (n = 3, mg/g).

Drying Method	Bioactive Compounds						
	Salidroside	Ayringin	Echinacoisde	Forsythoside B	Verbascoside	Isoacteoside	Oleuropein
SHD	2.471 ± 0.049 ^a^	0.149 ± 0.015 ^b^	11.734 ± 0.360 ^b^	11.351 ± 0.218 ^b^	4.486 ± 0.097 ^b^	0.077 ± 0.001 ^a^	21.490 ± 0.242 ^c^
OD	2.478 ± 0.072 ^a^	0.161 ± 0.005 ^b^	9.555 ± 0.056 ^d^	11.983 ± 0.040 ^a^	5.073 ± 0.022 ^a^	0.079 ± 0.001 ^a^	21.401 ± 0.117 ^c^
MD	2.646 ± 0.172 ^a^	0.028 ± 0.004 ^c^	10.422 ± 0.138 ^c^	9.256 ± 0.137 ^c^	2.817 ± 0.009 ^d^	0.052 ± 0.007 ^b^	31.735 ± 0.310 ^a^
SD	1.903 ± 0.062 ^c^	0.317 ± 0.040 ^a^	13.824 ± 0.121 ^a^	9.472 ± 0.170 ^c^	2.924 ± 0.030 ^c^	0.073 ± 0.003 ^a^	25.968 ± 0.261 ^b^
IRD	2.212 ± 0.070 ^b^	0.018 ± 0.007 ^c^	4.737 ± 0.189 ^e^	5.891 ± 0.058 ^d^	2.041 ± 0.016 ^e^	0.050 ± 0.017 ^b^	14.674 ± 0.038 ^d^

Note: SHD: shade-drying; OD: oven-drying; MD: microwave-drying; SD: sun-drying; IRD: infrared-drying. Values in the same column with different letters are significantly different (*p* < 0.05).

## Data Availability

All data are included in the manuscript in the form of figures and tables.

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
