# Peer review of "Influence of Five Drying Methods on Active Compound Contents and Bioactivities of Fresh Flowers from Syringa pubescens Turcz"

_molecules, 2023, doi:10.3390/molecules28237803_

Round 1

Reviewer 1 Report

Comments and Suggestions for Authors

The manuscript “ Influence of five drying methods on active compound contents and bioactivities of fresh flower from Syringa pubescens Turcz” submitted by Xu et al. Deals with the search for suitable drying processes to deal with this medicinal species, maintaning the antioxidant, anti-inflammatory and enzyme inhibitory activities. It is an intersting manuscript, with a good amount of work. I am sending some suggestions and recommend the authors to read the whole manuscript, as some minor mistakes, other than those described below, were identified.

Title: I suggest using flowers, in the plural.

Abstract:

Add the corresponding abbreviations after the drying methods (lines 14 and 15).

Since you wrote infrared, it should advisable using ultraviolet instead of UV.

Key words: Please, correct the plant name

Introduction:
Please, abbreviate the genus after the first citation: S. pubescens. (line 38)

Lines 51-53: Please, rephrase. It is confusing.

Lines 55-56: The phrase “The correlation between SPF bioactivity and 55 the drying method is lacking. ” repeats what was said just before. Please, remove this sentence.

Results and Discussion:

In lines 63 and 64, you cite the names of the compounds present in the plant. Please, provide the chemical structures for all of them.

Line 85. Please, define SEM

Line 98: Instead of “main chemical compounds of SPF samples ” I suggest using chemical profile, since FT-IR is not closely linked to the compound, but represents the chemical groups present. Therefore, if there are several minor compounds, but all of them have the same functional group, the IR signals can be related to the functional groups. In addition, (line 106), low bands are not only proportional to the compounds quantity, but also represent the absorption. Some bands are always low. I suggest to rephrase lines 104-107, taking this into consideration.

Lines 113-114: UV peaks cannot be used to identify compounds, theefore, I recommend to change the text for: “The peaks at 285 and 332 nm can be associated to the presence of oleuropein and phenylethanoid glyco sides, respectively [18, 19].”

In the same way, in line 114, I recommend this modification: It could be observed that the bands associated to the major compounds of ...”

Line 121: samples

Line 125: were

Figures 5 and 7 must be improved. It is not possible to read. Figure 8 can also be improved.

Line 331: Hydroxyl

Author Response

Dear Reviewer,

Thank you for your comments and suggestions on our manuscript titled “Influence of five drying methods on active compound contents and bioactivities of fresh flower from Syringa pubescens Turcz.” (Molecules-2678559). These comments helped us improve our manuscript, and provided important guidance for future research.

We have addressed the reviewer’s comments. And we hope this meets your requirements for a publication.

We marked the revised portions in red in the manuscript. And the main comments and our specific response are detailed below.

Reviewer 1#

The manuscript “Influence of five drying methods on active compound contents and bioactivities of fresh flower from Syringa pubescens Turcz” submitted by Xu et al. Deals with the search for suitable drying processes to deal with this medicinal species, maintaining the antioxidant, anti-inflammatory and enzyme inhibitory activities. It is an interesting manuscript, with a good amount of work. I am sending some suggestions and recommend the authors to read the whole manuscript, as some minor mistakes, other than those described below, were identified.

Q1: Title: I suggest using flowers, in the plural.

Response: Thank you for your careful review. The authors have revised this word “flowers”.

Abstract:

Q1: Add the corresponding abbreviations after the drying methods (lines 14 and 15).

Since you wrote infrared, it should advisable using ultraviolet instead of UV.

Response: The UV has been changed as ultraviolet.

Q2: Key words: Please, correct the plant name.

Response: Thank you for your careful comments. The authors have revised the plant name.

Introduction:

Q1: Please, abbreviate the genus after the first citation: S. pubescens. (line 38)

Response: The authors have revised the abbreviate.

Q2: Lines 51-53: Please, rephrase. It is confusing.

Response: Thank you for your insightful comments. The authors have revised the sentence. “Different drying method is suitable for different medicinal herbal because the drying process can lead to the loss of heat labile compounds”

Q3: Lines 55-56: The phrase “The correlation between SPF  bioactivity and 55 the drying method is lacking. ” repeats what was said just before. Please, remove this sentence.

Response: Thank you for your careful comments. The authors have deleted the sentence.

Results and Discussion:

Q1: In lines 63 and 64, you cite the names of the compounds present in the plant. Please, provide the chemical structures for all of them.

Response: Thank you for your valuable suggestions. The authors have added the Figure 1.

Q2: Line 85. Please, define SEM

Response: The authors have revised the SEM.

Q3: Line 98: Instead of “main chemical compounds of SPF samples ” I suggest using chemical profile, since FT-IR is not closely linked to the compound, but represents the chemical groups present. Therefore, if there are several minor compounds, but all of them have the same functional group, the IR signals can be related to the functional groups. In addition, (line 106), low bands are not only proportional to the compounds quantity, but also represent the absorption. Some bands are always low. I suggest to rephrase lines 104-107, taking this into consideration.

Response: Thank you for your valuable and insightful comments. The authors have changed the words using the chemical profile. Furthermore, according to the reviewer’s comments, the authors thought the results (lines 104-107) was not correct because some low bands could be caused by different reasons. Therefore, the section should be deleted.

Q4: Lines 113-114: UV peaks cannot be used to identify compounds, therefore, I recommend to change the text for: “The peaks at 285 and 332 nm can be associated to the presence of oleuropein and phenylethanoid glyco sides, respectively [18, 19].”

In the same way, in line 114, I recommend this modification: It could be observed that the bands associated to the major compounds of ...”

Response: Thank you for your valuable comments and suggestions. And the authors have revised according to the reviewer’s suggestions.  

Q5:

Line 121: samples

Line 125: were

Figures 5 and 7 must be improved. It is not possible to read. Figure 8 can also be improved.

Line 331: Hydroxyl

Response: The authors have revised according to the reviewer’s suggestions. And Figure 5, 7 and 8 have been changed using new figures.

Response: The authors have revised them.

Reviewer 2 Report

Comments and Suggestions for Authors

Development of novel drying methods and detailed their efficiency, furthermore, the optimization of conventional drying methods has high relevance not just for the science but also for the industry practice. Therefore, the topic of the manuscript can be considered as interesting. However, with present content, the manuscript needs significant revision and reconsidering before publishing 8see my comments).

Major comments:

Please give the temperatures/temperature profiles achieved by different drying methods. Temperature is considered as the main factor on the change of different bioactive components, and antioxidant properties, as well. Without of the temperature values the results cannot be considered as comparable.

In my opinion, the Abstract is too general. Please give some concretized results/data (for ’stronger radical scavenging ability’, ’higher reducing power’, ’ HD sample possessed the highest inhibitory activity’, etc.).

The Introduction section is too superficial. Please discuss the main characteristics, advantages-disadvantages, practical applicability and their effects on bioactive components for flowers, spices etc. in more details.

Please provide the details of the drying methods (power, frequencies, temperature control etc.). This part of the methodology section is very weak.

In my opinion, the process parameters of different drying methods should be optimised, and after compare the different methods.

Conclusion section is too superficial.

Minor comments:

Please reconsider the sentence in line 16 ’..chemicals differences were assessed using Scanning electron microscopy..’ .SEM can be used to investigate the morphology.

Results related to in vitro antioxidant activities (in section 2.5) are not discussed in details.

Figure 5 has poor visibility. please improve it (mainly the axis titles, scales, units, legends).

Author Response

Dear Reviewer,

Thank you for your comments and suggestions on our manuscript titled “Influence of five drying methods on active compound contents and bioactivities of fresh flower from Syringa pubescens Turcz.” (Molecules-2678559). These comments helped us improve our manuscript, and provided important guidance for future research.

We have addressed the reviewer’s comments. And we hope this meets your requirements for a publication.

We marked the revised portions in red in the manuscript. And the main comments and our specific response are detailed below.

Reviewer

Q1: Please give the temperatures/temperature profiles achieved by different drying methods. Temperature is considered as the main factor on the change of different bioactive components, and antioxidant properties, as well. Without of the temperature values the results cannot be considered as comparable.

Response: Thank you for your valuable and insightful comments. The authors have revised the description of different drying methods.

Q2: In my opinion, the Abstract is too general. Please give some concretized results/data (for ’stronger radical scavenging ability’, ’higher reducing power’, ’ HD sample possessed the highest inhibitory activity’, etc.).

Response: Thank you for your valuable suggestions. The authors have revised the abstract.

Q3: The Introduction section is too superficial. Please discuss the main characteristics, advantages-disadvantages, practical applicability and their effects on bioactive components for flowers, spices etc. in more details.

Response: Thank you for your valuable suggestions. The authors have added the description and discussion.

Q4: Please provide the details of the drying methods (power, frequencies, temperature control etc.). This part of the methodology section is very weak.

Response: Thank you for your valuable suggestions. The authors have added the description on the drying methods.

Q5: In my opinion, the process parameters of different drying methods should be optimised, and after compare the different methods.

Response: Thank you for your valuable and insightful suggestions and comments. In this manuscript, the drying methods were conducted according to the published literatures. The authors agree with the reviewer’s comments and suggestions, and the authors will investigate the process parameters including the MD and IRD in the future investigations.

Q6: Conclusion section is too superficial.

Response: The authors have revised the conclusion section.

Minor comments:

Q1: Please reconsider the sentence in line 16 ’..chemicals differences were assessed using Scanning electron microscopy..’ .SEM can be used to investigate the morphology.

Response: The authors have revised.

Q2: Results related to in vitro antioxidant activities (in section 2.5) are not discussed in details.

Response: The authors have added the discussion.

Q3: Figure 5 has poor visibility. please improve it (mainly the axis titles, scales, units, legends).

Response: The authors have changed the figures.

Round 2

Reviewer 2 Report

Comments and Suggestions for Authors

The authors have revised the manuscript thoroughly according to reviewers comments and suggestion. Rephrasings, amendments more detailed methodology and discussion made the manuscript more complete and clear.

Comments:

Please give the frequency of microwve irradition and IR radiation in the methodology section.

Please give the temperature profiles for every drying methods used in the research.

Please unify the texts size in Figure 2.

Please improve the visibility of axis titles and legends in Figure 5, 8 and Figure 9.

Author Response

Dear Reviewer,

Thank you for your valuable and insightful comments and suggestions on our manuscript again. We have addressed the reviewer’s comments. And we hope this meets your requirements for a publication.

We marked the revised portions in red in the manuscript. And the main comments and our specific response are detailed below.

Reviewer

Q1: Please give the frequency of microwave irradition and IR radiation in the methodology section.

Response: Thank you for your careful review. The authors have added the frequency of microwave irradition and IR radiation.

Q2: Please give the temperature profiles for every drying method used in the research.

Response: The temperature of SHD, OD and SD have been added in this manuscript. However, the temperature of MD and IRD could not be control. In this study, the temperature was not

Q3: Please unify the texts size in Figure 2.

Response: Thank you for your careful review. The authors have changed a new one.

Q4: Please improve the visibility of axis titles and legends in Figure 5, 8 and Figure 9.

Response: The authors have revised the axis titles and legends.